# General nonlinear Hall current in magnetic insulators beyond the quantum anomalous Hall effect

Daniel Kaplan [1], Tobias Holder [1] & Binghai Yan [1] ✉

Can a generic magnetic insulator exhibit a Hall current? The quantum anomalous Hall effect (QAHE) is one example of an insulating bulk carrying a quantized Hall conductivity while insulators with zero Chern number present zero Hall conductance in the linear response regime. Here, we find that a general magnetic insulator possesses a nonlinear Hall conductivity quadratic to the electric field if the system breaks inversion symmetry, which can be identified as a new type of multiferroic coupling. This conductivity originates from an induced orbital magnetization due to virtual interband transitions. We identify three contributions to the wavepacket motion, a velocity shift, a positional shift, and a Berry curvature renormalization. In contrast to the crystalline solid, we find that this nonlinear Hall conductivity vanishes for Landau levels of a 2D electron gas, indicating a fundamental difference between the QAHE and the integer quantum Hall effect.

Understanding electric conduction of insulators is fundamental to condensed matter physics. For example, the quantum Hall effect is a unique realization of a 2D topological insulating phase of matter, with distinct experimental signatures[1–4], notably a quantized Hall conductance $\sigma^{xy}$ which adheres to the quantized value $\frac{e^2}{h}$ with astonishing precision, up to at least $10^{-10}$ [5,6]. It has been known since the early days of the quantum Hall effect[7] that the quantization of $\sigma_{xy}$ is related to the Berry curvature in a periodic system, with the robustness of the quantization discussed in several works[8–10]. As a close cousin, the quantum anomalous Hall effect (QAHE) refers to the appearance of a quantized Hall conductivity in 2D systems even in the absence of a magnetic field[11–13]. First proposed by Haldane[14], the QAHE requires the breaking of time-reversal symmetry (TRS) in the crystal system characterized by a Chern number $C_N$ for the occupied bands. Consequently, a calculation at linear order yields for the Hall conductivity $\sigma^{xy} = C_N \frac{e^2}{h}$ [15,16]. The QAHE has been experimentally realized in several systems, notably magnetically doped thin films of topological insulators[17,18], stochiometric magnetic topological insulators[19], and recently in Moiré superlattices[20,21]. However, in contrast to the quantum Hall effect, careful experiments on the QAHE find a less precisely quantized Hall conductivity, with precision of 0.01%[18] and 0.1%[20], respectively.

While it is well established that the Hall conductivity is exactly quantized at linear order[7], we demonstrate that an intrinsic nonlinear conductivity can appear at finite bias for generic magnetic insulators. These nonlinear effects are due to virtual interband transitions, which have been shown to appear beyond linear response even in insulators[22,23]. The interband transitions induce changes to the magnetization and polarization of the material, which at nonlinear order lead to a finite Hall conductivity. An intuitive picture of these interband transitions is shown in Fig. 1: The quasiparticle response can be modified by shifts in velocity, shifts in position, and by a renormalization of the Berry curvature. These shifts constitute a nonlinear multiferroic response driven only by the applied electric field. Interestingly, the nonlinear correction to the Hall conductivity can be nonzero even if $C_N = 0$. In the following, we derive the nonlinear conductivity in quantum perturbation theory, and finally express our main result as a correction to the semiclassical equations of motion. One may wonder whether such effects lead to corrections to the integer quantum Hall effect. To this end, we show explicitly that in the case of a 2D electron gas, for both quadratic and linear dispersion, no corrections appear. The reason is that the Berry phase is entirely created by the applied magnetic field and thus extrinsic and independent of the underlying band structure, which is fully renormalized and transformed into Landau levels.

[1]Department of Condensed Matter Physics, Weizmann Institute of Science, Rehovot 7610001, Israel. ✉e-mail: binghai.yan@weizmann.ac.il

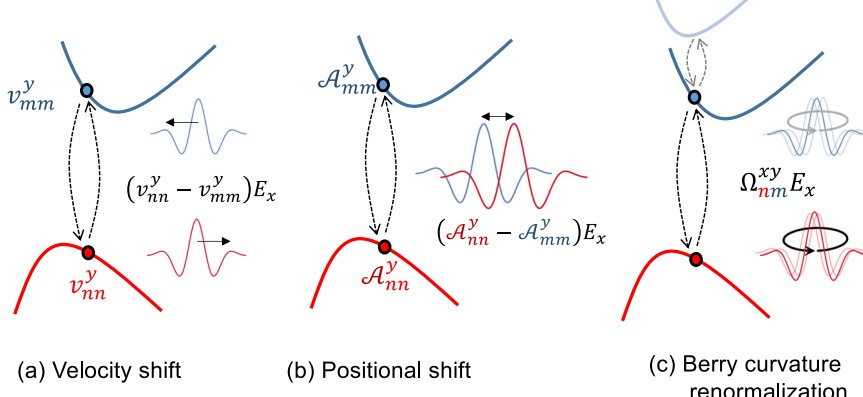

**Fig. 1 | Illustration of corrections to the AHE conductivity in the presence of an electric field ($E_x$). a** Velocity shift of the wave packet due to the transition between the occupied ($n$) and unoccupied ($m$) bands. **b** Positional shift of a wavepacket due to interband transitions. **c** Berry curvature renormalization by the third bands. All three contributions (see Eqs. (4), (5), and (6)), which are linear in $E_x$, are non-zero for a generic multiband dispersion which breaks both inversion and time-reversal symmetries.

## Results

### Theory

As is well known, an applied perturbation does not usually commute with the band Hamiltonian $H_0$ and will induce interband transitions in terms of the unperturbed eigenvalues of $H_0$. Thus, a wavepacket initially centered on a single Bloch periodic state $|W\rangle = \int d\mathbf{k} a_n(\mathbf{k})|n\mathbf{k}\rangle$ at $t = 0$, will evolve to be a linear combination containing contributions of many bands[24,25]. In the Kubo formalism this is reflected in the appearance of resonant contributions which are broadened by the finite quasiparticle lifetime $\tau$. We consider an insulator with broken inversion and time-reversal symmetries, i.e. it holds for the dispersion $\varepsilon_n(\mathbf{k}) \neq \varepsilon_n(-\mathbf{k})$. The uniform ($\mathbf{q} \to 0$) electric field $\mathbf{E} = \mathbf{E_0}e^{i\omega t}$ is introduced via its vector potential $\mathbf{A}(t) = \frac{\mathbf{E_0}e^{i\omega t}}{i\omega}$. By minimal coupling, the Bloch-periodic Hamiltonian transforms as $H_0(\mathbf{k}) \to H_0(\mathbf{k} - e\mathbf{A})$. The current operator is given by $J^c = \frac{\delta H}{\delta A}$. Up to $\mathbf{A}^2$ this yields

$$J^c = -ev^c + e^2 \sum_a \mathbf{A}^a w^{ac} - \frac{e^3}{2} \sum_{a,b} \mathbf{A}^a \mathbf{A}^b u^{abc}, \quad (1)$$

where $v^a = \partial_a H_0$, $w^{ab} = \partial_a \partial_b H_0$, $u^{abc} = \partial_a \partial_b \partial_c H_0$ and $\partial_a = \frac{\partial}{\partial k_a}$. The evaluation of the total current is then carried out using a Green's function approach[26,27],

$$\langle j^c \rangle = -i \int \frac{d^d k}{(2\pi)^d} \int \frac{d\Omega}{2\pi} \mathrm{Tr}(J^c A(\mathbf{k},\Omega)). \quad (2)$$

Here $A$ is $A(\mathbf{k},\Omega) = i(G_R - G_A)$ and $G_R$ and $G_A$ are the retarded and advanced Green's functions, respectively[27]. The the spectral function $A(\mathbf{k},\Omega)$ is found through a solution to the Dyson equation, giving $A(\mathbf{k},\Omega) = (1 + G^r \Sigma^r)A_0(1 + G^a \Sigma^a)$, and $G^r = G_0^r(1 + \Sigma^r G^r), G^a = G_0^a(1 + \Sigma^a G^a)$. In the usual manner[26], Dyson's equations are solved perturbatively. Since the electromagnetic coupling is Hermitian, $\Sigma^r = \Sigma^a = -\sum f \mathbf{A}^c(\omega)$, and $G^{r/a} = G_0^{r/a} \sum_{n=0}^{\infty} (\Sigma^{r/a} G_0^{r/a})^n$, and correspondingly $A(\mathbf{k},\Omega) = \sum_{n=0}^{\infty} (\Sigma^r G_0^r)^n G_0 \sum_{m=0}^{\infty} (\Sigma^a G_0^a)^m$. The diagrammatic expansion of $A(\mathbf{k},\Omega)$ has recently been developed yielding the complete response at 2nd order in $\mathbf{A}$[28,29]. In the Bloch basis $|n\mathbf{k}\rangle$ the unperturbed Green's functions are $G_{0,nm}^r(\Omega) = \frac{\delta_{nm}}{\Omega - \varepsilon_n + it^{-1}}, G_{0,nm}^a(\Omega) = \frac{\delta_{nm}}{\Omega - \varepsilon_n - it^{-1}}, A_{0,nm}(\Omega) = 2\pi i \delta_{nm} f_n \delta(\Omega - \varepsilon_n)$. Here $f_n$ is the Fermi occupation factor. We begin by considering $\mathbf{A}(\omega)$ at finite frequency, and then taking the limit $\omega \to 0$ (note that bold-face $\mathbf{A}(\omega)$ denotes the external, classical gauge field). Crucially, the $\omega \to 0$ pole is avoided by retardation in the form of

$\omega \to \omega + \frac{i}{\tau}$. The expansion of $A(\mathbf{k},\Omega)$ will contain a pole in the sum of frequencies, which is shifted by $\omega + (-\omega) \to \omega + (-\omega) + \frac{2i}{\tau}$. The result is then evaluated in the $\tau \to \infty$ limit. The expansion for the lifetime-free ($\tau^0$) contribution is detailed in the Supplementary Information. The full expansion for all orders of $\tau$ and the general expressions are presented in ref. 30. For concreteness, we present the case $\mathbf{E_0} = (E_x, 0, 0)$ and focus on two-dimensional systems. We stress that here we calculate the dc-component ($\omega \to 0$) of the nonlinear response tensor. At order $\tau^0$, and up to order $\mathcal{O}(\mathbf{A}^2)$ the transverse conductivity $\sigma^{xy}$ reads,

$$\sigma^{xy} = \frac{e^2}{\hbar} \sum_{n \in \text{occ.}} \int \frac{d^2 k}{(2\pi)^2} \left[ \Omega_{nn}^{xy} + eE_x(I_1 + I_2 + I_3)_{nn}^{xy} \right] \quad (3)$$

$$(I_1)_{nn}^{xy} = \left[\varepsilon^{-2} \mathcal{A}^x, \Delta^y \mathcal{A}^x\right]_{nn} - \left[\varepsilon^{-2} \mathcal{A}^y, \Delta^x \mathcal{A}^x\right]_{nn} \quad (4)$$

$$(I_2)_{nn}^{xy} = 2\left[\varepsilon^{-1} \mathcal{A}^x, S^{xy}\right]_{nn} - 2\left[\varepsilon^{-1} \mathcal{A}^y, S^{xx}\right]_{nn} \quad (5)$$

$$(I_3)_{nn}^{xy} = i\left[\varepsilon^{-1} \mathcal{A}^x, [\mathcal{A}^x, \mathcal{A}^y]\right]_{nn}. \quad (6)$$

Here, we introduced compact notation: $[A,B]_{nm} = \sum_{l \neq n,m} A_{nl} B_{lm} - (B \leftrightarrow A)$, and $\Delta_{nm}^{x,y} = v_{nn}^{x,y} - v_{mm}^{x,y}$, which is resolved using the Hadamard product, i.e., $(A\Delta^{x,y})_{nm} = A_{nm} \Delta_{nm}^{x,y}$. $\varepsilon_n(\mathbf{k})$ is the energy of the n-th Bloch band, at momentum $\mathbf{k}$, and is also inserted in the expression in the Hadamard form. $\mathcal{A}^{x,y}$ is the non-Abelian Berry connection, defined as usual $\langle n|\hat{\mathbf{r}}|m\rangle = \mathcal{A}_{nm}$, where $\mathbf{r}_{nm}$ is the position operator. $\varepsilon^{-1}$ appears in the commutators of Eqs. (4)–(6) should be read as energy differences. A complete expansion of the commutators is found in the SI. It is important to note that past theoretical treatments of the nonlinear Hall effect were often carried out in the Boltzmann approach. In this method, the electric field perturbs the Fermi-Dirac density of each band, individually. Operator corrections, diagonal in the band basis (such as the anomalous velocity), have to be introduced manually. In contrast, the quantum perturbative formalism used in the present work captures corrections to the current vertex, electron density, and dispersion simultaneously. We stress that the nonlinear Hall effect we derive here is distinct from the usual mechanisms discussed previously in the literature: it survives in the limit of the electric field frequency $\omega \to 0$, and does not require a Fermi surface, unlike known sources of nonlinear Hall signals (as predicted in refs. 31–33).

In writing Eq. (3), we split the nonlinear conductivity into three physically distinguishable response types. Namely, $I_1$ is associated with a velocity shift, $\Delta_{nm}^{\alpha}$, while $I_3$ describes a renormalization of the Berry curvature. Each of these is individually gauge invariant (see SI), and is the result of residual processes from the optical, high-frequency limit. The velocity shift $I_1$ has a form similar to the injection current seen in TRS-broken systems at high frequencies[29,34]. $I_3$ is a purely multi-band object seen at higher-order response. In the case of a coupling between magnetic and electric fields, it is related to the non-topological part of the magneto-electric polarizability[35]. Finally, $I_2$ involves a tensor $S^{xy}$ which is related to the shift vector found in optical response[36]. This quantity is defined as,

$$S_{nm}^{\alpha\beta} = (1 - \delta_{nm})\left(\lambda_{nm}^{\alpha\beta} - \frac{i}{2}\left(\mathcal{A}_{nm}^{\alpha}\delta_{nm}^{\beta} + \mathcal{A}_{nm}^{\beta}\delta_{nm}^{\alpha}\right)\right), \tag{7}$$

$$\lambda_{nm}^{\alpha\beta} = \frac{i}{2}\left(\left\langle n|\partial_\alpha\partial_\beta m\right\rangle - \left\langle \partial_\alpha\partial_\beta n|m\right\rangle\right). \tag{8}$$

$\lambda^{\alpha\beta}$ presents a higher derivative on the wavefunction, which results from the resolution of $\partial_\alpha\mathcal{A}_{nm}^{\beta}$. $\delta_{nn}^{\alpha} = \mathcal{A}_{nn}^{\alpha} - \mathcal{A}_{mm}^{\alpha}$ which encodes a real-space shift of the wave-function center[37] also appears, with the latter entering via a Hadamard product, thus rendering $S^{\alpha\beta}$ manifestly gauge covariant: $S_{nm}^{\alpha\beta} \to e^{i\theta_{nm}(\mathbf{k})}S_{nm}^{\alpha\beta}$, under the $U(1)^N$ gauge transformation, with the Bloch wavefunctions transforming as $|\psi_{n\mathbf{k}}\rangle \to e^{i\theta_n(\mathbf{k})}|\psi_{n\mathbf{k}}\rangle$. The commutator structure ensures the gauge invariance of the entire expression for $\sigma^{xy}$. A detailed proof of the gauge invariance under a $U(1)^N$ transformation for each of the terms is presented in the SI. The appearance of $\Delta^{x,y}$ as well as $\delta^{x,y}$ in Eq. (6) shows the connection of these objects to expressions at finite frequency such as injection and shift currents[29,38–40]. The second-order correction in Eq. (6) has several noteworthy properties:

### Absence of longitudinal components

The correction may be nonzero only in the direction perpendicular to the applied field and only enters the transverse components of $\sigma^{\alpha\beta}$ (in any dimension). This is of course due to the fact that the correction is related to the Berry curvature, which ensures that the resultant current is always perpendicular to the perturbation. Consequently, the correction does not violate charge conservation nor does it produce a longitudinal response which would require a finite Fermi surface[41].

**Multiband nature.** Inspection of Eqs. (4)–(6) reveals that the in-gap conductivity is generated by interband processes, which are due to virtual transitions between occupied and unoccupied bands. This is a direct result of the commutator structure because $\sum_n f_n[A, B]_{nn} = \sum_{n \neq m} f_{nm} A_{nm} B_{mn}$ where $f_{nm} = f_n - f_m$ (See SI for further details). The latter vanishes if both states are occupied or empty. Since $\Omega_{nn}^z$ can be projected into a single-band, it appears at linear order. But corrections to this are manifestly multi-band objects, involving direct probes of the states through $\Delta_{nm}^{\alpha}, \varepsilon_{nm}, S_{nm}^{\alpha\beta}$. The presence of the shift tensor $S_{nm}^{\alpha\beta}$ suggests a property which is encoded in at least two bands, as the commutator in which it appears restricts $n \neq m$. Furthermore, we note the presence of a higher-order multi-band term, $[\mathcal{A}^x\varepsilon^{-1}, \Omega^z]_{nn}$, which is $(I_3)_{nn}$ in Eq. (6). To parse this object, one evaluates $\Omega_{nm}^z$, where $n \neq m$. Using the definition of the commutator, however, $[A, B]_{nm} = \sum_{l \neq n,m} A_{nl}B_{lm} - (A \leftrightarrow B)$. From this, it follows that this term only exists for three bands or more. In the two-band limit, the commutator can be directly evaluated to be $[A, B]_{12} = \sum_{l \neq 1,2} A_{1l}B_{l2} - (A \leftrightarrow B) = 0$, as the sum cannot extend over *any* intermediate state. This term represents, therefore, a unique signature of a quantum process which involves interband transitions between two principle bands – occupied and empty – with an assisting interim third band.

## Symmetries

The correction to the anomalous Hall conductivity strongly depends on the underlying symmetry of the crystal lattice. Firstly, a general requirement for the appearance of intrinsic in-gap responses is the breaking of TRS. Since this effect is quadratic in the electric field, inversion symmetry ($P$) must be broken as well. The symmetry discussion is simplified by considering the correction as a *second-order* Hall conductivity. We define $\sigma^{xx;y} = \frac{\delta^2 j_y}{\delta E_x \delta E_x}$. Eqs. (4)–(6) show that the Hall part of the tensor $\sigma^{ab;c}$ takes the form $\sigma^{aa;c}$. Applying the von Neumann principle[42], we find that for all rotational symmetries $C_{n,z}$, $n \geq 2$, $\sigma^{aa;c}$ vanishes identically. In 3D (or higher), other components of the response tensor are permitted, e.g., of the form $\sigma^{xx;z}$. The emergence of a longitudinal in-gap current is restricted by the presence of point-group symmetries. In the case of $C_{3z}$, for example, $\sigma^{xx;y} = -\sigma^{yyy}$, but since the correction vanishes for $\sigma^{yyy}$, the Hall response is null as well.

## Twisted bilayer graphene

As a candidate system to test our results, we suggest using strained twisted bilayer graphene (TBG)[43–46]. As previously seen in the case of resonant optical conductivity[36], in TBG second-order electrical responses can become exceptionally large due to the large phase space for transitions between flat bands. We model a time-reversal breaking state of TBG by considering the Bistrizer-MacDonald[43] continuum model for a single valley and spin of TBG at a twist angle of $\theta = 1.05°$. This corresponds to experimental measurements of TBG in the ferromagnetic, 3/4 filled state[20,47], in a series of cascading symmetry-broken states in this system[48,49]. This phase is topological with a Chern number $C_N = 1$. In a sample, the TBG is usually placed on top of a layer of hBN[50,51], which breaks inversion symmetry. This can be modeled by introducing a staggered potential $\Delta = 17\text{meV}$[46]. Since TBG on top of hBN still retains a $C_{3z}$-symmetry, the correction considered here remains zero. This is shown in Fig. 2b. Figure 2c also reveals that the momentum distribution of the correction is antisymmetric within the mini Brillouin zone (mBZ) and thus vanishes after integration. However, introducing strain breaks $C_{3z}$, rendering the correction in Eq. (3) nonzero, as seen in Fig. 2d, with the mBZ modified as well. The deviation from $\sigma_0^{xy} = \frac{e^2}{h}$ increases with increasing strain. Using a typical strain amplitude of $\epsilon \sim 0.65\%$[52], and electric field strengths of $E = 300\,\text{V}\,\text{m}^{-1}$, it reaches a value of 0.1%, which is comparable to the deviation from perfect quantization in recent experiments[20].

## Semiclassical interpretation

The structure of the correction permits the following semiclassical form. We define the electric field-induced shift tensors,

$$v_E^a = e\frac{\mathcal{A}^a\boldsymbol{\Delta}^b}{\varepsilon}E_b, \quad S_E^a = e\boldsymbol{S}^{ab}E_b, \quad \Omega_E = e\Omega^{ab}E_b, \tag{9}$$

Where all terms enter as Hadamard products. In-band basis, these objects are translated as $v_E^a = \sum_b \frac{e}{\varepsilon_{nm}}\mathcal{A}_{nm}^a\Delta_{nm}^b E_b$. This can be carried out analogously for all terms. The semi-classical anomalous current at second order can then be written as

$$\mathbf{j} = \frac{e^2}{h}\sum_{n\in\text{occ.}}\mathbf{E}\times\int\frac{\text{d}^2k}{(2\pi)^2}\left(\boldsymbol{\Omega}_{nn} + \left[\frac{\mathcal{A}}{\varepsilon}\times\mathsf{V}\right]_{nn}\right). \tag{10}$$

Here, $\mathsf{V}_E^a = v_E^a + S_E^a + \Omega_E^a$. The cross product is to be interpreted as usual[15,53], such that $\left[\frac{\mathcal{A}}{\varepsilon}\times\mathsf{V}\right]_{nn} = \sum_{m\in\text{unocc.}}\epsilon^{abc}\frac{\mathcal{A}_{nm}^b}{\varepsilon_{nm}}\mathsf{V}_{mn,E}^c - (n\leftrightarrow m)$. Here $\epsilon^{abc}$ is the Levi-Civita symbol. This partition into three pieces is identical in content to the previous decomposition into $I_1, I_2, I_3$ in Eq. (3). $\frac{\mathsf{V}^a}{h}$ carries units of velocity, meaning that it is the velocity of the

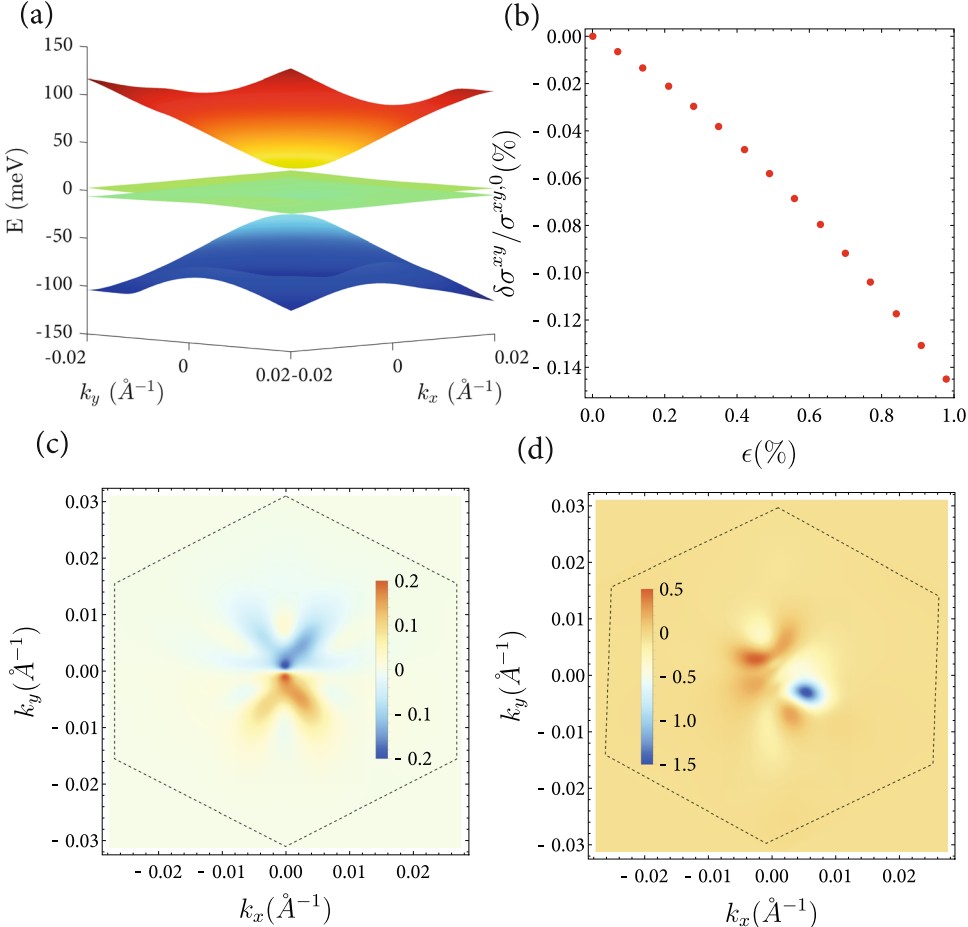

**Fig. 2 | Nonlinear correction to the anomalous Hall in TBG. a** Band structure of twisted bilayer graphene for $\theta \sim 1.05°$. Remote dispersive bands contribute to the interband correction, which vanishes in the single-band limit. **b** Magnitude of the correction to the AHC $\sigma^{xy,0} = \frac{e^2}{h}$ as a function of applied uniaxial strain $\epsilon$ in TBG. Momentum space distribution of $\delta\sigma^{xy}$ for $\epsilon = 0$ (**c**) and $\epsilon = 0.8\%$ in (**d**).

(instantaneous) charge displacement upon application of the external electric field **E**. At first order, this displacement modifies the position operator through a change in the charge dipole. At second order in the applied field, this deformation couples back to the position operator, resulting in a correction to the anomalous velocity, now effectively quadratic in the applied field. The weight $\varepsilon_{nm}^{-1}$ attached to the position operator reflects the quantum-perturbative expansion, since the $V^a$ is now explicitly *inter-band*, and the transitions to the neighboring bands are suppressed by the energy gap. The nonlinear correction to the Hall conductivity thus vanishes when all unoccupied bands are infinitely separated from the top of the valence band such that $\varepsilon_{nm} \to \infty$. A visualization of the momentum-space structure of Re(V) using a simplified two band model can be found in the SI.

**Robustness of the integer quantum Hall effect**

The precise quantization of the conductivity $\sigma^y$ of the integer quantum hall effect for a 2D electron gas can be understood in the absence of any higher-order corrections at finite bias. To show that our correction vanishes identically for Landau levels, consider the Hamiltonian of an electron gas in the Landau gauge, $\mathbf{A} = (0, -Bx, 0)$,

$$H = \frac{p_{x'}^2}{2M} + \frac{M\omega_c^2 x'^2}{2}. \tag{11}$$

Here as usual $\omega_c = \frac{eB}{M}$, and $x' = x + \frac{\hbar k_y}{eB}$. In the presence of the gauge field **A** the kinetic momenta become $p \to p + e\mathbf{A} = \pi$. We shall show that

the quantization of the Hall conductivity is guaranteed by the ladder operator structure. The velocity matrix elements in the Landau level basis are,

$$v_x = \frac{\partial H}{\partial \pi_{x'}} = \frac{p_{x'}}{M} = i\sqrt{\frac{\hbar\omega_c}{2M}}(a^\dagger - a), \tag{12}$$

$$v_y = \frac{\partial H}{\partial \pi_y} = \omega_c x' = \frac{\omega_c l_B}{\sqrt{2}}(a + a^\dagger). \tag{13}$$

We define the ladder operators $a = \frac{1}{\sqrt{2m\hbar\omega_c}}(ip_{x'} + M\omega_c x')$ and $l_B = \sqrt{\frac{\hbar}{eB}}$. For Landau levels, $\varepsilon_n = \hbar\omega_c(n + 1/2)$, $a^\dagger|n\rangle = \sqrt{n+1}|n+1\rangle$, and $a|n\rangle = \sqrt{n}|n-1\rangle$. The expectation values become $\langle n|v_x|m\rangle = i\frac{\hbar}{\sqrt{2}Ml_B}\left(\sqrt{m+1}\delta_{n,m+1} - \sqrt{m}\delta_{n,m-1}\right)$, $\langle n|v_y|m\rangle = \frac{\omega_c l_B}{\sqrt{2}}\left(\sqrt{m+1}\delta_{n,m+1} + \sqrt{m}\delta_{n,m-1}\right)$. The quantization of the linear conductivity is directly related to the ladder structure of operator algebra in the integer quantum Hall fluid. A demonstration of this property is relegated to the SM. However, the fact that the ladder operators only connect Landau levels with energy differences $\Delta\varepsilon = \pm\hbar\omega_c$ can be used to show that *all* higher-order corrections vanish for the 2D electron gas at high magnetic field. At 2nd order, the relevant diagrams of the quantum perturbative calculation give two contributions at order $\tau^0$ (all other terms vanish in the gapped phase identically)[30]. For the Hall response

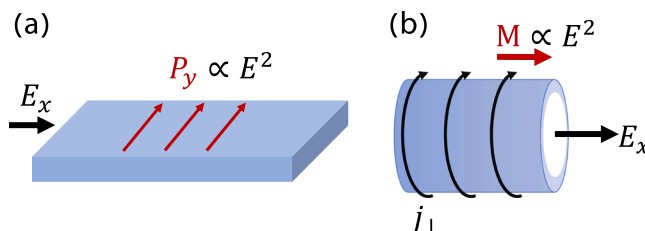

**Fig. 3 | Multiferroic character of the nonlinear response. a** In a sample with physical edges, a net polarization perpendicular to the applied field accumulates at the edge. **b** When the geometry is cylindrical, the application of an electric field produces a perpendicular current, thereby inducing a magnetization. Since there are no edges, no polarization can accumulate. In both scenarios, $P_y, M \propto E^2$.

tensor $\sigma^{xx,y}$,

$$\sigma^{xx,y} = \frac{e^3}{\hbar^2} \sum_n f_n \left[ -2 \left[ \varepsilon^{-3} w^{xx}, v_y \right]_{nn} - \left[ \varepsilon^{-3} v^x, w^{xy} \right]_{nn} \right] + \frac{e^3}{\hbar^2} \sum_{n,m} \left[ -4 \frac{f_{nm}}{\varepsilon_{nm} \varepsilon_{nl}^3} v^x_{nm} v^x_{ml} v^y_{ln} \right.$$
$$\left. -2 \frac{f_{nm}}{\varepsilon_{nm}^2 \varepsilon_{nl}^2} v^x_{nm} v^x_{ml} v^y_{ln} - \frac{f_{nm}}{\varepsilon_{nm}^3 \varepsilon_{nl}} v^x_{nm} v^x_{ml} v^y_{ln} \right].$$

(14)

The elimination of the first two commutators in Eq. (14) is due to the free fermion dispersion of the Landau levels giving $w^{ab}_{nm} = \langle n| \frac{\partial^2 H}{\partial \pi_a \partial \pi_b} |m\rangle \propto \delta_{nm}$ since the underlying dispersion is quadratic in Eq. (11). The commutator $[A, B]_{nn}$ contains only off-diagonal components of $A_{nm}, B_{nm}$. Consequently, since $w^{ab}_{nm} = 0$ for any $a, b, n \neq m$ this contribution vanishes. We are left with the triple product $v^x_{nm} v^x_{ml} v^y_{ln}$. By applying the ladder structure for $v^{x,y}_{nm}$ the following combinations appears: $\delta_{n,m\pm1}\delta_{m,l\pm1}\delta_{l,n\pm1}$ (The two-band part of this formula trivially vanishes, because neither vx nor vy have any diagonal terms in the incompressible phase.). By applying, e.g., the middle Kronecker delta, we have the condition that $n = l \pm 1 \pm 1$, and $n = l \mp 1$. Clearly, there exists no $l, n$ that satisfies this constraint. This results in $\sigma^{xx,y} = 0$ regardless of the exact structure of the Hamiltonian, provided the algebra of the ladder operators is preserved. To further stress this point, we can carry out an analogous calculation on a system with Dirac dispersion, containing a non-trivial Berry phase. In this case the Hamiltonian is $H = \hbar v_f((k_x + e\mathbf{A}_x)\sigma_x + (k_y + e\mathbf{A}_y)\sigma_y + \Delta\sigma_z)$, where we included $\Delta\sigma_z$ for band inversion and a finite Berry curvature. $\sigma_{x,y,z}$ are Pauli matrices. For this dispersion, the operator $w^{ab} = 0$ generally, since $\frac{\partial^2 H}{\partial \pi_a \partial \pi_b} = 0$ by construction. Following Hunt et al.[54], the eigenstates are spinors of the form $\psi_n = (\psi_{n-1}, \psi_n)$. The velocity operators are constants and are given by $v_{x/y} = \hbar v_f \sigma_{x,y}$. Once again, $v_{x/y,nm} \propto \delta_{n,m\pm1}$, due to the fact that the Pauli matrices $\sigma_x, \sigma_y$ pair spinor components with the quantum number $n$ differing by $\pm 1$. Importantly, the diagonal component $v_{x/y,nn} = 0$, as long as $\Delta$ is momentum independent. The generalization of the above can be made by considering that $n$-th order response will contain an $n+1$ product of velocity operators $v^x_{m_1 m_2} v^x_{m_2 m_3} v^x_{m_3 m_4} \cdots v^y_{m_n m_1}$, which produces the condition that $\delta_{n_1 n_2 \pm 1} \delta_{n_2 n_3 \pm 1} \delta_{n_3 n_4 \pm 1} \cdots$ which yields zero for the real part of the current at any order. The only nonzero combination for which band indices can be selected appears at order $n = 1$ corresponding to linear response, which gives the quantized integer Hall conductivity.

## Multiferroic response

The electric field dependent magnetization change we have derived, as shown in Eq. (10) should be viewed as a magnetization induced by the electric field. This is evidenced by the fact that the term $\frac{e}{\hbar} \left[ \frac{A}{\varepsilon} \times V \right] d^2 k$ carries precisely the units of magnetization density and its additive nature with respect to the Berry curvature. The coupling of this term at

2nd order in the electric field to the Hall current also generates a polarization density $\mathbf{p} \approx \mathbf{j}\tau$, where $\tau$ is the characteristic relaxation time in the system. Similarly, the requirement of low symmetry in the system (and inversion symmetry breaking) points to a generalization of multiferroics[55,56] to the case of an electric field inducing changes both to the magnetization and the polarization, albeit at higher order. This is to be contrasted with the traditional multiferroic picture, in which changes to either magnetization or polarization occur *linearly* with the applied electric or magnetic fields, respectively. Eq. (10) is the first demonstration of a new type of multiferroic response, which occurs uniquely at nonlinear order. We further note that the induced effects here are dissipationless as the power $\mathbf{j} \cdot \mathbf{E} = 0$. An illustration of this process is presented in Fig. 3. The instantaneous current $V^a$ produces a magnetization similar to the classical mechanism via $\mathbf{r} \times \mathbf{j}$. Here $\mathbf{r}$ is represented by the Berry connection $\mathcal{A}$ and due to the effect being driven by inter-band transitions, the position operator is weighted by the interband coherence factor $(\varepsilon_n - \varepsilon_m)^{-1}$. In a system with physical edges (Fig. 3(a)), the effect will be manifested in an accumulated polarization. If the planar sample is folded on itself, creating a hollow cylinder (Fig. 3b), no polarization will accumulate and a net magnetization will be generated. In both cases, the total change is proportional to $\propto \mathbf{E}^2$, introducing a nonlinear multiferroic response.

## Discussion

We have shown that in general magnetic insulators which break inversion, time reversal as well as rotational symmetries, a quadratic correction to the in-gap Hall conductivity appears. In a topological phase, this indicates that measurements at finite bias will deviate from the quantized value due to the presence of nonlinear corrections. As an example, we calculated the correction for strained twisted bilayer graphene, finding for the magnitude of the nonlinearity values which are comparable with the observed precision of the quantization in the recent experiment of ref. 20. Another experimental signature may appear in the non-reciprocal nature of the conductivity. Namely, in systems where the correction is observable, we find that $\sigma^{xy} \neq -\sigma^{yx}$, and the sum $\sigma^{xy} + \sigma^{yx}$ can thus be treated as a proxy for the correction. Thirdly, the quantities derived here might be visible as non-linear powers in the $I-V$ curve.

We note that the nonlinear Hall conductivity is finite even when the Chern number vanishes in a magnetic insulator. This raises interesting questions about the possible boundary states in such a system, which is subject to future studies. As a way to understand this result in a bulk picture, one can imagine a Corbino geometry in which edge states are absent. In such a scenario without physical edges, charge transport has been predicted to occur through bulk spectral flow[57]. The QAHE in a Corbino disk has recently been observed experimentally[58].

Recent progress on nonlinearities in graphene superlattices[59] suggests that experiments at moderate finite bias on graphene-based systems are possible. By tuning the graphene superlattices to the QAH state and sweeping the bias, the nonlinear corrections, as well as the non-reciprocity they produce might be accessible. In addition, the sensitivity of the effect to strain suggests an electro-mechanical setup in which a controlled application of tensile stress is employed in order to modify the Hall conductivity (at finite bias). We note that systems with $C_{3z}$ symmetry, such as doped $Bi_2Se_3$[17] do not exhibit this correction due to the symmetry restriction. However, new material platforms, such as transition-metal dichalcogenide superlattices[21] are promising avenues for the investigation of the nonlinear QAHE, with tunability by external knobs such as a displacement field. Indeed, ref. 21 observed deviations from exact quantization in $\sigma^{xy}$. Our results might be relevant in understanding why experiments on systems with rotational symmetries observe a much more precisely quantized QAHE[18,60]. Related to that, the reasoning presented here raises the question whether third or even higher-order corrections are non-

vanishing even if a QAH system has inversion and $C_3$ symmetry. The nonlinear QAHE establishes a concrete difference in the quantization of the QAHE compared to the IQHE, which suggests that using QAHE systems for metrology depends on subtleties related to the crystal systems, symmetries, and the magnitude of the applied bias. Our results predict a striking phenomenon that a generic insulator can present a nonlinear current response in the dc limit.

## Data availability

All data needed to evaluate the conclusions in the paper are present in the paper and/or the Supplementary Information. Additional data related to this paper may be requested from the authors.

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

## Acknowledgements

We thank Ady Stern and Xi Dai for useful discussions. B.Y. acknowledges the financial support from the European Research Council (ERC Consolidator Grant No. 815869, "NonlinearTopo") and Israel Science Foundation (ISF No. 2932/21). D.K. appreciates support from the Weizmann Institute Sustainability and Energy Research Initiative.

## Author contributions

B.Y. conceived and supervised the research. D.K. derived the conductivities and numerically evaluated them for the TBG model. All authors analyzed the data and wrote the manuscript.

## Competing interests

The authors declare no competing interests.
