## [Peer Review File · Nature Communications]

General nonlinear Hall current in magnetic insulators beyond the quantum anomalous Hall effectREVIEWER COMMENTS

Reviewer #1 (Remarks to the Author):

The authors have developed a general theory for the second-order Hall response for generic insulator phases with the breaking of both time reversal and inversion symmetries. Their derivation is based on the non-linear perturbation theory leading to an expansion with respect to the scattering time τ . The most interesting result is that they found a τ -independent term for the clean limit, which provides a next-order correction to the Hall conductance in systems with quantized linear Hall conductance. These results are certainly new and important if they are correct. However, the physical meaning of such a nonlinear current response is not very clearly illustrated in the paper. Here, I have the following puzzles.

1) recently, the nonlinear Hall effect has been widely studied for metallic systems. The authors should explicitly explain in the paper that the effect they proposed in this paper is quite different from the ordinary nonlinear Hall effect. The latter is an AC effect and is for metallic systems. Whereas the present effect survives in the DC limit and it is for insulators.

2) Does this effect rely on the existence of the chiral edge modes? The authors claimed that such an effect commonly exist for all magnetic insulators without inversion symmetry. But the material example provided in the paper is the TBG systems with nonzero Chern number and the quantum Hall systems with the Landau levels. Both systems contain chiral edge modes.

3) If such an effect can exist for normal magnetic insulators with $C=0$ and it is obviously non-dissipative, it should be explained by the modification of the diamagnetic current of the magnetic insulator generated by the electric field. Can such an effect be also explained as a special type of "multi-ferroic" response, where the change of magnetization can be induced by the electric field?

Reviewer #2 (Remarks to the Author):

The manuscript under review studies nonlinear Hall effect in magnetic insulators. The second-order conductivity, which is the focus of the paper, is generically nonzero in systems with broken inversion symmetry. The authors find that in magnetic (time-reversal breaking) insulators the second-order Hall conductivity is finite provided certain conditions are satisfied. In particular, they find that corresponding correction can be split into three physically different terms, originating from velocity shift, position shift, and the renormalization of Berry curvature. Furthermore, the symmetry analysis performed in the paper shows that this correction is nonzero only if all the rotational symmetries are broken. Finally, the authors applied their results to strained twisted bilayer graphene and made comparison with integer quantum Hall effect.

While the results presented in the paper are interesting, they rely significantly on the previous works in the area of nonlinear Hall effect and are too specialized for the broad readership of Nature Communications. Furthermore, the way this paper is written seems rather technical, hence I find it more suitable for a specialized journal, at least in its current form. I would also appreciate if authors made more connection to the previous literature in this field. In particular, intrinsic nonlinear Hall effect was studied in a number of works, e.g., Phys. Rev. Lett. 127, 277201 (2021), arXiv:2208.02972, Phys. Rev. Lett. 112, 166601 (2014). However, if I understand correctly, in all those works finite density of states (i.e., metallic regime) was required for a nonzero result. Did those works overlook something important, or they just focused on a different setup, e.g., assuming some symmetries which forbid nonlinear Hall response in the insulating regime? Furthermore, I have a number of less significant comments and questions to the authors:

1. The second-order response to an external perturbation at a finite (even if small) frequency is either dc response or has doubled frequency. Which response do you calculate, dc or second

harmonic?

2. Below Eq. (2), you mention that the diagrammatic expansion of lesser Green's function ($G^<$) was developed in Refs. [29,30]. If I understand those papers correctly, they don't specifically use Keldysh formalism, unlike Ref. [61], which does.

3. Some notations and definitions must be corrected and clarified below Eqs. (4)-(6). In particular, indices in the definition of $[A,B]_{\{nm\}}$ should be fixed (index l is missing now). Does using the Hadamard form imply that terms like $\epsilon^{-1}A^x$ only have diagonal nonzero components (since ϵ^{-1} is diagonal)? Or one should read these epsilons as energy differences? Please clarify. Also, I would recommend the authors to add some more details on the derivation of Eqs. (4)-(6) to the supplement, since these are crucial equations.

4. In section "Multiband nature", you emphasize that the corrections are manifestly multiband. Is 2 considered as multi, or one needs $n > 2$? If 2 is enough, then I believe it is usually possible to express the answers in terms of the Abelian (diagonal) geometric objects, e.g., Ω_{nn} . Furthermore, you say that "Since Ω_{nn} can be projected into a single-band, it appears at linear order. But corrections to this are manifestly multi-band objects,.." I would like to stress that in order to have nonzero Berry curvature Ω_{nn} multiple bands are required anyway, even though Ω_{nn} itself can be attributed to (defined within) a given single band.

5. In the same section, there's formula $f_n [A,B]_{\{nn\}} = f_{\{nm\}}A_{\{nm\}}B_{\{mn\}}$. It seems like some summation signs are missing here.

6. In section "Symmetries", you find that the Hall response you study is nonzero only provided all the rotational symmetries are broken. Is it important for this result that the system is insulating, and if so, why exactly?

7. You apply your results to strained bilayer graphene. However, you don't present any details of the corresponding calculations. It would be very useful for nonexperts in TBG if you added a section in the supplement describing this calculation. It could at least contain the model you consider explicitly, with the discussion of the relevant symmetries which are broken.

8. Do you provide any details on the derivation of Eq. (10)?

9. In section "Semiclassical interpretation" you say that "The single band limit can BE (a typo) recovered in the limit where all unoccupied bands are infinitely separated from the top of the valence band..." Again, this statement is not quite correct. Even though the band gap may not appear explicitly in this limit, finite Berry connection necessarily requires the presence of multiple bands, even if they're well separated.

10. In sentence below Eq. (13) "the latter operators" should be changed to "the ladder operators".

11. Do I understand correctly that your calculation in section "Robustness of the integer quantum Hall effect" is only applicable for free Schrodinger particles, i.e., it cannot be applied to electrons in a crystal periodic potential or to Dirac electrons? Also, it seems you assume that an integer number of Landau levels is filled and there's no partially filled level. Why do you assume that? Wouldn't that require finite tuning of density and/or magnetic field?

12. Finally, supplementary information should be cleaned up and multiple typos corrected, if you intend to submit it along with your paper. For example, a sentence in the first paragraph says, "In the diagrammatic picture, 4 diagrams contribute (may be insert these later)." I believe the text in parenthesis was not intended for a reader. And yes, I believe the diagrams should be included as well. Another example is "The nonlinearities L_x, L_y do not merely break inversion and mirror symmetries remaining mirror but add higher..." Either it's quite confusing or there's a typo (I mean double "mirror"). Finally, the caption to Fig. S1 says ", the correction decays like M^{-1} ...". I believe it should be M^{-1} instead. I think there're even more typos, so I'd recommend the authors to go over their supplement carefully.

Reviewer #3 (Remarks to the Author):

In this manuscript, the authors discuss the quantization of the Hall coefficient in the quantized anomalous Hall effect (QAHE). Typically, the QAHE is associated with the perfect quantization of the induced current in the direction orthogonal to an applied electric field in the absence of an external magnetic field. In this manuscript, the authors examine the subleading terms in the electric field E which means that the Hall coefficient quantization is no longer exact at finite E . The authors apply their theory to a recent experiment in twisted bilayer graphene (Ref 20) and find that the order of magnitude of the correction is around 0.1%, in agreement with the experimental measurements. Finally, the authors argue that this correction is a concrete difference between the quantization of the QAHE and IQHE.

The calculations appear sound, although I have not verified all the technical details involved in the derivations. I found some parts of the main text overly technical (e.g. the section below Eq 2, and discussion of IQHE), but they do not detract from the overall point, which is quite clear. The result could be significant if the arguments for its applicability to recent experiments on QAHE can be made stronger. I also have some questions regarding some of the authors claims.

On the robustness of the integer quantum Hall effect: it is shown that there is no correction to the Hall coefficient for the integer quantum Hall effect, as opposed to the QAHE discussed previously. This is shown for the 2D electron gas in a magnetic field.

I find this example to be rather artificial, since the 2DEG is quite idealized. Since the authors' goal is to identify fundamental differences between IQHE and QAHE, it would make sense to consider also IQHE arising from other types of dispersion. For example, a Dirac dispersion or something with non-zero Berry curvature. At present, this section feels a bit odd, since the 2DEG in a magnetic field is exactly solvable and the quantization of the Hall coefficient is known. I suppose the 2DEG result serves as a nice consistency check of the authors formalism, but beyond that I am not convinced of its relevance.

The authors claim that their theory explains the small non-quantization observed in twisted bilayer graphene in Ref. 20. I am not yet convinced of this as the experimental extraction of the Hall coefficient in Ref 20 involves additional details.

1. the experimental measurement is done at finite frequency (as opposed to the zero frequency limit discussed in the manuscript).
2. the Hall resistivity R_{xy} in Ref 20 is extracted via a "symmetrization method" where measurements at positive and negative B field are combined. Is the higher order correction proposed by the authors expected to survive this symmetrization?
3. Is 300V/m a realistic electric field strength for this particular experiment?

Finally, I wonder if the authors can comment on the QAHE recently observed in moire transition metal dichalcogenides [Nature 600, 641–646 (2021)]. In this experiment, there appears to be a small (few percent) correction to the quantized anomalous Hall coefficient. Could the slight non-quantization be explained by the authors theory?

Typos: After Eq 6, there is a typo in the definition of the commutator (the index l does not appear in the expression).

In the paragraph under "Multiband nature", $(a \leftrightarrow B)$ should be $(A \leftrightarrow B)$.

1 Ref. 1

Reviewer #1 (Remarks to the Author):

The authors have developed a general theory for the second-order Hall response for generic insulator phases with the breaking of both time reversal and inversion symmetries. Their derivation is based on the non-linear perturbation theory leading to an expansion with respect to the scattering time τ . The most interesting result is that they found a τ -independent term for the clean limit, which provides a next-order correction to the Hall conductance in systems with quantized linear Hall conductance. These results are certainly new and important if they are correct. However, the physical meaning of such a nonlinear current response is not very clearly illustrated in the paper. Here, I have the following puzzles.

We are thankful to the referee for highlighting the novelty and importance of our results.

1) recently, the nonlinear Hall effect has been widely studied for metallic systems. The authors should explicitly explain in the paper that the effect they proposed in this paper is quite different from the ordinary nonlinear Hall effect. The latter is an AC effect and is for metallic systems. Whereas the present effect survives in the DC limit and it is for insulators.

We agree with the referee that our proposed effect survives in the dc-limit and is different from the usual ac effect studied in **metallic** systems. We have added the following clarification in main text,

We stress that the novel nonlinear Hall effect we derive here is distinct from the usual mechanisms discussed previously in the literature: it survives in the limit of the electric field frequency $\omega \rightarrow 0$, and does not require a Fermi surface, unlike known sources of nonlinear Hall signals.

2) Does this effect rely on the existence of the chiral edge modes? The authors claimed that such an effect commonly exist for all magnetic insulators without inversion symmetry. But the material example provided in the paper is the TBG systems with nonzero Chern number and the quantum Hall systems with the Landau levels. Both systems contain chiral edge modes.

We appreciate the referee's important clarification. In the supplementary material, we show that a simple two band model with nonlinearities with $C = 0$ will host the nonlinear anomalous Hall effect we have shown here. The effect of topological bands is reflected in a weaker decay of our effect i. e. instead of decaying like $M^{-3/2}$ as a function of the gap size, it only decays according to M^{-1} in the topological phase, as expected from naive scaling considerations.

A simple way to understand that our results *do not* rely on chiral edge modes is through the lens of the linear quantum anomalous Hall effect. To this end, consider the linear anomalous Hall effect in the Corbino geometry, as recently realized in the experiment of Kawamura *et al.* (Nat. Phys 2023, <https://doi.org/10.1038/s41567-022-01888-2>). In the Corbino geometry (as theoretically argued by König *et al.*, Phys. Rev. B 90, 165435 (2014)), bulk charge transport cannot be explained by edge states due to there being no physical edges. Instead, Laughlin-like pumping arguments are employed to show that transport occurs by spectral flow through the bulk. If charge pumping is considered beyond linear order in the varying flux $\partial_t \Phi$, nonlinear corrections necessarily appear, which is equivalent to the mechanism in this manuscript. In a planar geometry involving physical edges the same arguments apply for a bulk calculation.

One might anticipate additional effects that appear at nonlinear order, such as charge accumulation. Our calculation is insensitive towards effects like these, and also induced edge states or the modification of existing edge states due to the applied electric field. However, these effects are physically distinct from the change to the bulk Hall response at nonlinear order. These effects are interesting for future study.

To further stress this point, we have added a clarifying paragraph to the main text:

We note that the nonlinear Hall conductivity is finite even when the Chern number vanishes in a magnetic insulator. This triggers interesting questions about the possible boundary states in such a system, which is subject to future studies. As a way to understand this result in a bulk picture, one can imagine a Corbino geometry in which edge states are absent. In such a scenario without physical edges, charge transport has been predicted to occur through bulk spectral flow König (2014). The Corbino disk was recently realized experimentally Kawamura (2023).

3) If such an effect can exist for normal magnetic insulators with $C=0$ and it is obviously non-dissipative, it should be explained by the modification of the diamagnetic current of the magnetic insulator generated by the electric field. Can such an effect be also explained as a special type of “multi-ferroic” response, where the change of magnetization can be induced by the electric field?

We deeply appreciate this comment by the referee. This is precisely the meaning behind our semiclassical treatment of this nonlinear Hall effect. Eqs. 9-10 describe an induced current, which we parametrize via $\frac{V^a}{\hbar}$ – the instantaneous electric field-induced velocity. In turn, the object generating the current is a magnetization, given by $[\frac{A}{\varepsilon}, V^a]$, in agreement with the multiferroic interpretation given by the referee. The first order correction found in V^a can be thought-of as an instantaneous change to the magnetization, that in turn induces a polarization at 2nd order.

The fact that the effect appears in low symmetry systems is further evidence of a multi-ferroic response. It is well known (Physics 2, 20 (2009)) that a general realization of magnetization/polarization switching in a multi-ferroic system requires lowered

symmetry in the crystal. To further enhance this point and incorporate the referee's suggested connection between multiferroics and the novel nonlinear response we propose, we have added a section in the introduction which reads,

These nonlinear effects are due to virtual interband transitions, which have been shown to appear beyond linear response even in insulators Michishita (2021), Kaplan (2020). The interband transitions induce changes to the magnetization and polarization of the material, which at nonlinear order lead to a finite Hall conductivity.

We further introduce the subject of new multiferroic effects in a new section, with a new figure, Fig. 3, that illustrates our new multiferroic mechanism.

The electric field dependent magnetization change we have derived, as shown in Eq. 10 should be viewed as a magnetization induced by the electric field. This is evidenced by the fact that the term $\frac{e}{\hbar} \left[\frac{\mathbf{A}}{\varepsilon} \times \mathbf{V} \right] d^2k$ carries precisely the units of magnetization density and its additive nature with respect to the Berry curvature. The coupling of this term at 2nd order in the electric field to the Hall current also generates a polarization density $\mathbf{p} \approx \mathbf{j}\tau$, where τ is the characteristic relaxation time in the system. Similarly, the requirement of low symmetry in the system (and inversion symmetry breaking) points to a generalization of multiferroics Khomskii (2009), Spaldin (2019), to the case of an electric field inducing changes both to the magnetization and the polarization, albeit at higher order. This is to be contrasted with the traditional multiferroic picture, in which changes to either magnetization or polarization occur *linearly* with the applied electric or magnetic fields, respectively. Eq. 10 is the first demonstration of a new type of multiferroic response, which occurs uniquely at nonlinear order. We further note that the induced effects here are dissipationless as the power $\mathbf{j} \cdot \mathbf{E} = 0$

We once more thank the referee for his observations regarding the novelty and importance of our work, and we are glad to incorporate their insightful suggestions and comments.

2 Ref. 2

The manuscript under review studies nonlinear Hall effect in magnetic insulators. The second-order conductivity, which is the focus of the paper, is generically nonzero in systems with broken inversion symmetry. The authors find that in magnetic (time-reversal breaking) insulators the second-order Hall conductivity is finite provided certain conditions are satisfied. In particular, they find that corresponding correction can be split into three physically different terms, originating from velocity shift, position shift, and the renormalization of Berry curvature. Furthermore, the symmetry analysis performed in the paper shows that this correction is nonzero only if all the rotational symmetries are broken. Finally, the authors applied their results to strained twisted bilayer graphene and made comparison with integer quantum Hall effect.

We appreciate the referee's succinct characterization of our main results.

While the results presented in the paper are interesting, they rely significantly on the previous works in the area of nonlinear Hall effect and are too specialized for the broad readership of Nature Communications. Furthermore, the way this paper is written seems rather technical, hence I find it more suitable for a specialized journal, at least in its current form.

Nat. Commun., has recently published detailed overviews of nonlinear Hall effects (for example: Nat Commun 10, 3047 (2019), or Nat Commun. 12, 5038 (2021)). These works are somewhat technical, and rely quite a bit on prior knowledge of nonlinear Hall literature. Yet they have immediately attracted wide appreciation in the community, and rightly so. Compared to these works, the present manuscript does not nearly as much rely on subject-specific knowledge.

Regardless, it is in our great interest to make the manuscript as broadly accessible as possible. Therefore, to address the referee's criticism, we have now made the introduction more general, and less focused on the matter of quantization. Additionally, we made sure that the derivation does not require previous knowledge about the formalism of (non)linear response. We have also modified the supplementary information, where additional explanations are supplied for all the relevant expressions that are used in the main text, and the way they are to be calculated in the Bloch band basis.

I would also appreciate if authors made more connection to the previous literature in this field. In particular, intrinsic nonlinear Hall effect was studied in a number of works, e.g., Phys. Rev. Lett. 127, 277201 (2021), arXiv:2208.02972, Phys. Rev. Lett. 112, 166601 (2014). However, if I understand correctly, in all those works finite density of states (i.e., metallic regime) was required for a nonzero result.

In the revised manuscript, we now refer to the previous works the referee has pointed out. We also comment specifically that our derived nonlinear Hall differs dramatically from previous results, in that it does not rely on a finite density of state at the chemical

potential.

Did those works overlook something important, or they just focused on a different setup, e.g., assuming some symmetries which forbid nonlinear Hall response in the insulating regime?

We are grateful for this question, which pinpoints one of our main advances regarding anomalous Hall responses: We would like to specifically stress two important distinctions separating our work from the past:

(1) The semiclassical formalism, used in the referee's suggested papers (Phys. Rev. Lett. 112, 166601 (2014), Phys. Rev. Lett. 127, 277201 (2021)), considers the distribution function of one band, in isolation. Finite lifetimes are then introduced only as a relaxation mechanism for out-of-equilibrium shifts to the electronic density. All field-related corrections to operators are introduced manually, and only in band-diagonal form. Conversely, our quantum perturbative approach captures all the effects stemming from *nonlinear* corrections to current operators, band broadening, and energy as well as density corrections. Only when all these effects are taken into account, can our proposed nonlinear Hall effect be recovered.

(2) As the referee correctly points out, and as we explain in the introduction of the main text, in systems with rotational symmetries, the nonlinear response that we find indeed vanishes. Therefore, the majority of model systems considered in previous theory works cannot capture this novel nonlinear Hall effect.

In answering the referee's remark here, we realize that this point bears further clarification. We have now added the following paragraph in the introduction,

It is important to note that past theoretical treatments of the nonlinear Hall effect was often carried out in the Boltzmann approach. In this method, the electric field perturbs the Fermi-Dirac density of each band, individually. Operator corrections, diagonal in the band basis (such as the anomalous velocity), have to be introduced manually. In contrast, the quantum perturbative formalism used in the present work, captures corrections to the current vertex, electron density and dispersion simultaneously.

Furthermore, I have a number of less significant comments and questions to the authors:

1. The second-order response to an external perturbation at a finite (even if small) frequency is either dc response or has doubled frequency. Which response do you calculate, dc or second harmonic?

We thank the referee for this question. We apologize it was not made clearer. We are calculating the dc response. This is now stressed above the start of the derivations section.

2. Below Eq. (2), you mention that the diagrammatic expansion of lesser Green's function ($G^<$) was developed in Refs. [29,30]. If I understand those papers correctly, they don't specifically use Keldysh formalism, unlike Ref. [61], which does.

We agree and concur with the referee. Our intention here was to introduce the conventional notation used in standard derivations of currents in interacting and disordered systems (for example, we followed the conventions introduced in R. Jishi “Feynman Diagram Techniques in Condensed Matter Physics”, 2013) . We regret that this led to some confusion regarding the technique employed in the present work. To clarify, we have constructed a nonlinear Kubo formula. To resolve any ambiguity, we replaced $G^<$ with the symbol $A(\mathbf{k}, \Omega)$ which denotes the spectral function. This spectral function is by definition given by $A(\mathbf{k}, \Omega) = i(G_R - G_A)$, where G_R and G_A are the retarded and advanced Green’s functions, respectively. The expansion of the Dyson equation proceeds as written in the main text.

3. Some notations and definitions must be corrected and clarified below Eqs. (4)-(6). In particular, indices in the definition of $[A, B]_{nm}$ should be fixed (index l is missing now). Does using the Hadamard form imply that terms like $\epsilon^{-1}A^x$ only have diagonal nonzero components (since ϵ^{-1} is diagonal)? Or one should read these epsilons as energy differences? Please clarify. Also, I would recommend the authors to add some more details on the derivation of Eqs. (4)-(6) to the supplement, since these are crucial equations.

The Hadamard form used here indicates an entry-wise product of two matrices Δ_{nm} and \mathcal{A}^x , Therefore, $(\Delta\mathcal{A}^x)_{nm} = \Delta_{nm}\mathcal{A}_{nm}^x$. ϵ should be read only as energy differences, with the band indices indicated by the underlying operator it is multiplying. For example, $(\epsilon^{-1}\mathcal{A}^x)_{nm} = \frac{\mathcal{A}_{nm}^x}{\epsilon_{nm}} = \frac{\mathcal{A}_{nm}^x}{\epsilon_n - \epsilon_m}$. We have added a sentence explaining the role of ϵ^{-1} and created a section in the SI describing the exact parsing of all the terms as they appear in the main text for further clarity.

4. In section “Multiband nature”, you emphasize that the corrections are manifestly multiband. Is 2 considered as multi, or one needs $n > 2$? If 2 is enough, then I believe it is usually possible to express the answers in terms of the Abelian (diagonal) geometric objects, e.g., Ω_{nn} . Furthermore, you say that “Since Ω_{nn} can be projected into a single-band, it appears at linear order. But corrections to this are manifestly multi-band objects,..” I would like to stress that in order to have nonzero Berry curvature Ω_{nn} multiple bands are required anyway, even though Ω_{nn} itself can be attributed to (defined within) a given single band.

We thank the referee for this question and observation. Indeed, as we show in the SI, the first two corrections I_1, I_2 (Eqs. (4)-(5)) are finite even for two bands. The Berry curvature renormalization, however, requires more than 2 bands, $n > 2$. We underline this point more clearly in the revised version. To avoid any confusion, we also removed the sentence pointed out by the referee concerning the Berry curvature.

5. In the same section, there’s formula $f_n[A, B]_{nn} = f_{nm}A_{nm}B_{mn}$. It seems like some summation signs are missing here.

We have added the relevant summation symbol, such that this equation now reads $\sum_n f_n[A, B]_{nn} = \sum_{n \neq m} f_{nm} A_{nm} B_{mn}$

6. In section “Symmetries”, you find that the Hall response you study is nonzero only provided all the rotational symmetries are broken. Is it important for this result that the system is insulating, and if so, why exactly?

We thank the referee for this question. This touches at the core of our findings. Indeed, in a situation where all symmetries are lifted, it is possible that terms which depend on a finite density of states at the Fermi level contribute to the conductivity. We would like to highlight two important differences, however: 1. In the gapless phase, the conductivities obtained in Phys. Rev. Lett. Phys. Rev. Lett. 112, 166601 (2014), Phys. Rev. Lett. 127, 277201 (2021), arXiv:2208.02972 are expected to dominate any inter-band corrections, as they are weighted by a factor related to the density of states at the Fermi level (delta function). 2. If the system is insulating, the Fermi surface contributions vanish. Therefore, tuning the system to the insulating state is a prerequisite to isolating the corrections found in this work.

Therefore, we work specifically in the insulating state where previously found contributions to the Hall conductivity will be zero due to the vanishing density of states. The gapped regime is the only in which transport is dominated by interband transitions induced by band broadening, as predicted in the present work.

7. You apply your results to strained bilayer graphene. However, you don’t present any details of the corresponding calculations. It would be very useful for nonexperts in TBG if you added a section in the supplement describing this calculation. It could at least contain the model you consider explicitly, with the discussion of the relevant symmetries which are broken.

Our numerical treatment of TBG with strain is based on a modified form of the Bistrizer-Macdonald (B-M) model, introduced previously in great detail in Phys. Rev. Research 4, 013209 (2022) and Nat. Commun. 11, 1650 (2020). We understand the referee’s concern and request for more detail. In the revised SI we have added a section describing the model, and comment about the specific symmetries which are broken. Within the B-M model, inversion symmetry is broken through a sublattice potential Δ (assumed to be due to alignment with hBN). Time reversal symmetry breaking is modelled by filling 3 out of the 4 Moiré bands. C_{3z} symmetry, which is found in the B-M model by construction, is broken by applying uniaxial strain.

8. Do you provide any details on the derivation of Eq. (10)?

Below Eq. 10, we provided a succinct recipe for the calculation of $[\frac{A}{\epsilon} \times \mathbf{V}]$. We understand however, that this may have been insufficient. This equation is obtained by re-expressing the quantities $I_1 - I_3$, such that the symplectic structure of commutators is replaced by the cross-product. Such a transformation is possible only in 2D. We com-

ment on this in the SI.

Eq. 10 is obtained by considering the equilibrium current, valid up to first order in \mathbf{E} (Rev. Mod. Phys. 82, 1959, Eq. (3.6) therein), where,

$$\mathbf{j} = \sum_n \int f_n \left(\frac{\partial}{\partial \mathbf{k}} \varepsilon_n + \mathbf{E} \times \boldsymbol{\Omega}_n \right) d^2k. \quad (1)$$

As we illustrated in Q6 of the referee, in the gapped phase, the contribution of the classical velocity $\frac{\partial \varepsilon}{\partial \mathbf{k}}$ vanishes, because it is a total derivative over the full BZ. As explained in the Sec. *Absence of longitudinal components*, the derived current is purely Hall-like. Hence, in 2D it can be written as an anti symmetric contraction of the electric field with itself, via the second order conductivity. In other words, \mathbf{j} must be of the form $\mathbf{j} \sim \mathbf{E} \times (\sigma \mathbf{E})$. We use this to pull out the leading $\mathbf{E} \times \dots$, as it multiplies both the Berry curvature and $\sigma \mathbf{E}$. We then use this to add the correction $\sigma \mathbf{E}$ directly to $\boldsymbol{\Omega}_{nn}$.

9. In section “Semiclassical interpretation” you say that “The single band limit can BE (a typo) recovered in the limit where all unoccupied bands are infinitely separated from the top of the valence band...” Again, this statement is not quite correct. Even though the band gap may not appear explicitly in this limit, finite Berry connection necessarily requires the presence of multiple bands, even if they’re well separated.

We thank the referee for this comment. We agree that our phrasing was not accurate enough. In the revised version, this sentence specifically reflects that the **nonlinear corrections** vanish in the limit where all the bands are infinitely separated from the ground state band n .

10. In sentence below Eq. (13) “the latter operators” should be changed to “the ladder operators”.

We have corrected this typo.

11. Do I understand correctly that your calculation in section “Robustness of the integer quantum Hall effect” is only applicable for free Schrodinger particles, i.e., it cannot be applied to electrons in a crystal periodic potential or to Dirac electrons? Also, it seems you assume that an integer number of Landau levels is filled and there’s no partially filled level. Why do you assume that? Wouldn’t that require finite tuning of density and/or magnetic field?

In partially filled Landau levels, metallic states may indeed emerge (c.f. Phys. Rev. B. 93, 085110). In such a scenario, there might be a finite density of states at the Fermi level. Thus one would have additional Fermi surface-like terms affecting the nonlinear conductivity, as pointed out previously in Phys. Rev. Lett. 112, 166601 (2014), Phys. Rev. Lett. 127, 277201 (2021). The scenario in which the experimental signature distinguishes our results most obviously from previous work is realized in the gapped phase. We further note that the integer quantum Hall (IQH) gapped state is amenable

to a relatively simple theoretical description, in terms of Landau levels, which is what made the calculation tractable. We also wish to highlight the theoretical implications of our results: the protection of the quantization of the Hall conductance, inherent to a gapped IQH state, is different from a gapped periodic system on a torus.

Concerning the referee's very valid comment about the experimental realization of our set-up *Wouldn't that require finite tuning of density and/or magnetic field?*, we are certain that by working in the condition outlined, e.g., in Rev. Mod. Phys. 58, 519 (1986), the integer quantum Hall phase – which will **not** experience corrections to the Hall conductivity – corroborating our theory, can be accessed experimentally.

12. Finally, supplementary information should be cleaned up and multiple typos corrected, if you intend to submit it along with your paper. For example, a sentence in the first paragraph says, "In the diagrammatic picture, 4 diagrams contribute (may be insert these later)." I believe the text in parenthesis was not intended for a reader. And yes, I believe the diagrams should be included as well. Another example is "The nonlinearities L_x, L_y do not merely break inversion and mirror symmetries remaining mirror but add higher. . ." Either it's quite confusing or there's a typo (I mean double "mirror"). Finally, the caption to Fig. S1 says ", the correction decays like $M - 1$. . . ". I believe it should be M^{-1} instead. I think there're even more typos, so I'd recommend the authors to go over their supplement carefully.

We are thankful to the referee for raising this point. Unfortunately, during the submission process, we attached an incomplete form of the supplementary information. In the resubmitted version, we have now attached the correct version of the SI. Importantly, the new SI also includes the referee's desired clarifications and corrections: we added a section which expands on the way mathematical objects appearing in the main text are derived and parsed; we included a section detailing the model used for strained TBG. Finally, we corrected all the typos the referee has pointed out, and combed for others ourselves. We also included the diagrams relevant for the calculation of the conductivity and explained how lifetimes are treatment during the evaluation of the terms.

With the help of the referee, we believe we have made our manuscript more accessible, corrected and clarified several points, elaborated on the details of the numerical calculations, and added additional information regarding the mathematical form of the conductivity. We would like to sincerely thank the referee for their comments, questions and suggestions. We have implemented their recommendations fully and believe the resubmission should meet with their approval.

3 Ref. 3

In this manuscript, the authors discuss the quantization of the Hall coefficient in the quantized anomalous Hall effect (QAHE). Typically, the QAHE is associated with the perfect quantization of the induced current in the direction orthogonal to an applied electric field in the absence of an external magnetic field. In this manuscript, the authors examine the subleading terms in the electric field E which means that the Hall coefficient quantization is no longer exact at finite E . The authors apply their theory to a recent experiment in twisted bilayer graphene (Ref 20) and find that the order of magnitude of the correction is around 0.1, in agreement with the experimental measurements. Finally, the authors argue that this correction is a concrete difference between the quantization of the QAHE and IQHE.

We thank the referee for this summary of our premise and main results.

The calculations appear sound, although I have not verified all the technical details involved in the derivations. I found some parts of the main text overly technical (e.g. the section below Eq 2, and discussion of IQHE), but they do not detract from the overall point, which is quite clear. The result could be significant if the arguments for its applicability to recent experiments on QAHE can be made stronger.

We thank the referee for their estimation of the validity of our results. It is of course our wish to make our results more relevant in terms of recent experiments on the QAHE, which we have done in the new version of our manuscript, with the referee's help.

I also have some questions regarding some of the authors claims. On the robustness of the integer quantum Hall effect: it is shown that there is no correction to the Hall coefficient for the integer quantum Hall effect, as opposed to the QAHE discussed previously. This is shown for the 2D electron gas in a magnetic field. I find this example to be rather artificial, since the 2DEG is quite idealized. Since the authors' goal is to identify fundamental differences between IQHE and QAHE, it would make sense to consider also IQHE arising from other types of dispersion. For example, a Dirac dispersion or something with non-zero Berry curvature. At present, this section feels a bit odd, since the 2DEG in a magnetic field is exactly solvable and the quantization of the Hall coefficient is known. I suppose the 2DEG result serves as a nice consistency check of the authors formalism, but beyond that I am not convinced of its relevance.

We thank the referee for raising this point. We begin by commenting that, as the referee correctly points out, whenever a new theoretical expression emerges it is important to cross-check it against long-standing, accepted limits. This was precisely our motivation in including in the section on the vanishing corrections for the 2DEG. We also agree with the referee that a quadratically dispersing electron gas model may be in some sense artificial. In the new version, we include results for a single gapped Dirac cone (which indeed has a non-trivial Berry phase); for the Dirac Landau levels the corrections are

again zero. We further stress and underline that this is precisely the consequence of the algebraic structure of Landau levels and their unique level spacing. This is precisely the mechanism that distinguishes them from the magnetic insulators whose momenta lie on the torus, giving instead non-zero corrections.

We would also like to point out another important facet of our results: seminal studies on the protection of the Hall conductivity σ^{xy} focused on two channels: disorder, and interactions (as an overview, we consider Phys Rev B 22 5142 (1980), Phys Rev B 46 2223 (1992)). To our knowledge, previous works did not discuss corrections induced due to *the external applied field*. The vanishing corrections, both for the quadratic 2DEG and the Dirac dispersion (in the new version) clearly show a new element of the protection in the quantum Hall phase: protection against field-induced corrections.

The authors claim that their theory explains the small non-quantization observed in twisted bilayer graphene in Ref. 20. I am not yet convinced of this as the experimental extraction of the Hall coefficient in Ref 20 involves additional details. 1. the experimental measurement is done at finite frequency (as opposed to the zero frequency limit discussed in the manuscript).

Following the supplementary data of Serlin *et al.* (Science Vol 367, Issue 6480 pp. 900-903, 2020)), the highest frequency used in the AC measurement is reported as $f = 5.5\text{Hz}$. We may therefore compare the typical momentum relaxation time in bilayer graphene with this frequency. The situation in which $\omega\tau \ll 1$ will allow us to treat the effective response as “dc-like”, since the frequency is substantially smaller than the relaxation rate. Available high-quality data for bilayer graphene (Phys. Rev. Lett. 98, 176805 (2007)) suggests that τ_p – the momentum relaxation time is $\tau_p \sim 0.15 \cdot 10^{-12}\text{s}$, which gives a frequency of $\tau_p^{-1} \sim 10^{13}\text{Hz} \gg f \sim 5\text{Hz}$. We are therefore confident that our dc calculation captures the experimental regime.

2. the Hall resistivity R_{xy} in Ref 20 is extracted via a ”symmetrization method” where measurements at positive and negative B field are combined. Is the higher order correction proposed by the authors expected to survive this symmetrization?

We thank the referee for this very important question. Indeed, symmetrization is an important tool in experiment for the acquisition of high quality data, and we wish to explain how our results are affected by this procedure. In Ref. 20, two symmetrizations techniques are invoked: 1. $R_{xy}(B) = \frac{R_{xy}(B) - R_{xy}(-B)}{2}$, (Eq. S2 in the Suppl. Info, *ibid*). This field symmetrization is **entirely compatible** with our expressions. We assume here that the field B controls the “direction” of the spontaneous time-reversal symmetry breaking. We underscore again that the conductivities I_1, I_2, I_3 in the main text are odd under the time-reversal operation. For example, considering I_1 and denoting the time-reversal action by T , $T^\dagger I_1 T = T^\dagger \sum_{nm} f_n \frac{A_{nm}^x A_{mn}^x \Delta_{mn}^y}{(\varepsilon_n - \varepsilon_m)^2} T = -f_n \frac{A_{nm}^x A_{mn}^x \Delta_{mn}^y}{(\varepsilon_n - \varepsilon_m)^2}$. This is so, because the only component that changes sign under time-reversal in this expression is $\Delta_{mn}^y = v_{mm}^y - v_{nn}^y$, as the velocity operator is odd under time-reversal

by definition. Analogously, all the conductivities I_1, I_2, I_3 transform in this manner. Therefore, following the symmetrization of Eq.S2 in Ref. 20, our terms acquire a factor of two (after integration over the full BZ) and survive. 2. The symmetrization carried out in Eq. S4 of Ref. 20 does not have a simple interpretation (in operator form) in the two-terminal paradigm in which we calculate our conductivities. In the concluding section of our manuscript, we specifically advocate for a different form of symmetrization: the measurement of $\sigma^{xy}(B) + \sigma^{yx}(B)$, taken at the **same** magnetic field. The form naturally cancels out the linear response component, since it must obey $\sigma_{\text{linear}}^{xy} = -\sigma_{\text{linear}}^{yx}$. Thus, a non-zero $\sigma^{xy}(B) + \sigma^{yx}(B)$ enables the observation of our predicted corrections.

3. Is 300V/m a realistic electric field strength for this particular experiment?

Indeed, in proposing this value, we tried to stay as close to experimental conditions as possible. We again refer to Ref. 20 for the source of this data point. In the main text, the authors estimate the relative length of their sample $L \approx 10^{-6}m$ while the current in the experiment is of the order $\sim 20\text{nA}$ (see Supplementary note S2 in Ref. 20). We take the reachable value of $I = 30\text{nA}$ which gives a current density of $j \sim 300\mu\text{Acm}^{-1}$. Using Fig. S4 (ibid), the two-terminal resistance is $\rho \sim 1\text{M}\Omega$ (see shaded yellow region where the anomalous Hall effect is observed). With these parameters in mind, we easily recover that, since $\rho j = E$, $E = 300\text{Vcm}^{-1}$ as indicated in the main text.

Finally, I wonder if the authors can comment on the QAHE recently observed in moire transition metal dichalcogenides [Nature 600, 641–646 (2021)]. In this experiment, there appears to be a small (few percent) correction to the quantized anomalous Hall coefficient. Could the slight non-quantization be explained by the authors theory?

We thank the referee for pointing out this work, and we indeed wish to make our work as closely related to existing experimental evidence, as possible. We have added a reference to this work, and we comment in the main text that such a platform is highly relevant for the future

We reiterate that the nonlinear correction proposed here is observable under the following conditions: in a 2D system, we require inversion and time reversal symmetry breaking, as well as breaking of all point-group symmetries (perpendicular mirrors, rotational symmetries).

Referring specifically to Nature 600, 641–646 (2021), we find that the conditions we indicated are generally fulfilled: the heterobilayer breaks inversion symmetry while the $\nu = 1$ band-inverted state indeed breaks time reversal symmetry. Rotational symmetry is also apparently broken, since the two lattices are incommensurate. A more detailed calculation is required to infer the magnitude of our corrections in this particular system, as one key difference from the TBG case is the need for a displacement field, for the band inversion. The Moiré physics of transition metal dichalcogenides are a platform for future work, where we will explore the effect of a displacement field on the nonlinear Hall effect.

Typos: After Eq 6, there is a typo in the definition of the commutator (the index l does not appear in the expression). In the paragraph under "Multiband nature", $(a \leftrightarrow B)$ should be $(A \leftrightarrow B)$

With the referee's help, we have fixed these typos throughout the main text and supplementary information.

We are grateful to the referee for their helpful comments, question and suggestions. We believe the comments improved the paper significantly.

REVIEWERS' COMMENTS

Reviewer #2 (Remarks to the Author):

I am in general satisfied with the authors' reply and the modified version of the manuscript. I recommend it for publication in Nature Communications in its current form.

Reviewer #3 (Remarks to the Author):

The authors have satisfactorily addressed the concerns of the referees, and I recommend publication.

1 Ref. 2

Reviewer #2 (Remarks to the Author):

I am in general satisfied with the authors' reply and the modified version of the manuscript.

I recommend it for publication in Nature Communications in its current form.

We thank the referee for his re-evaluation of our manuscript and we agree it is ready for publication in Nature Communications.

2 Ref. 3

Reviewer #3 (Remarks to the Author): The authors have satisfactorily addressed the concerns of the referees, and I recommend publication.

We thank the referee for his/her review and we agree that the manuscript is ready for publication.